# Injecting Background Knowledge into Embedding Models for Predictive Tasks on Knowledge Graphs

Claudia d'Amato[1,2], Nicola Flavio Quatraro[1], and Nicola Fanizzi[1,2]

[1] Dipartimento di Informatica – Università degli Studi di Bari Aldo Moro
n.quatraro@studenti.uniba.it
[2] CILA – Università degli Studi di Bari Aldo Moro
{claudia.damato,nicola.fanizzi}@uniba.it

**Abstract.** Embedding models have been successfully exploited for Knowledge Graph refinement. In these models, the data graph is projected into a low-dimensional space, in which *graph structural information* are preserved as much as possible, enabling an efficient computation of solutions. We propose a solution for injecting available background knowledge (schema axioms) to further improve the quality of the embeddings. The method has been applied to enhance existing models to produce embeddings that can encode knowledge that is not merely observed but rather derived by reasoning on the available axioms. An experimental evaluation on link prediction and triple classification tasks proves the improvement yielded implementing the proposed method over the original ones.

**Keywords:** Knowledge graphs · embeddings · link prediction · triple classification · representation learning

## 1 Introduction

Knowledge Graphs (KGs) are becoming important for several research fields. Although a standard shared definition for KGs is still not available, there is a general consensus on assuming them as organizations of data and information by means of graph structures [12]. KGs are often the result of a complex (integration) process involving multiple sources, human expert intervention and crowdsourcing. Several examples of large KGs exist, spanning from enterprise products, such as those built by Google and Amazon, to mention a few of them, to other open KGs such as the well known DBpedia, Freebase, Wikidata and YAGO [12]. Despite significant efforts for making KGs as comprehensive and reliable as possible, due to the complex building process they tend to suffer of two major problems: incompleteness and noise [20, 12]. As an example, it was found that about 70% of the persons described in DBpedia lack of information regarding their nationality and birth place [8]. Thus, a significant research effort have been devoted to *knowledge graph refinement*, aiming at correcting these issues with KGs [22]. Among the others, two tasks have gained a major attention: *Link Prediction*, aiming at predicting missing links between entities, and *Triple Classification*, that consists in assessing the correctness of a statement with respect to a KG.

In recent years, numeric approaches to these tasks have gained considerable popularity on account of their effectiveness and scalability when applied to large KGs.

Such models typically map entities and relations forming complex graph structures to simpler representations (feature-vectors) and aim at learning prediction functions to be exploited for the mentioned tasks. Particularly, the scalability purpose motivated the interest delved towards *embedding models* [4] which have been shown to ensure good performances on very large KGs. Knowledge graph embedding methods aim at converting the data graph into an optimal low-dimensional space in which *graph structural information* and *graph properties* are preserved as much as possible [4, 15]. The low-dimensional spaces enable computationally efficient solutions that scale better with the KG dimensions. Graph embedding methods differ in their main building blocks: the *representation space* (e.g. point-wise, complex, discrete, Gaussian, manifold), the *encoding model* (e.g. linear, factorization, neural models) and the *scoring function* (that can be based on distance, energy, semantic matching or other criteria) [15]. In general, the objective consists in learning embeddings such that the score of a valid (positive) triple is lower than the score of an invalid triple standing for a sort of negative examples. A major problem with these models is that KGs are mostly encoded exploiting available positive assertions (examples) whilst negative constraints are more rarely found, making negative examples more hardly derivable [2]. As positive-only learning settings may be tricky and prone to over-generalization, negative examples (invalid triples) have to be sought for either by randomly *corrupting* true/observed triples or deriving them having made the *local-closed world assumption* on the data collection. In both cases, wrong negative information may be generated and thus used when training and learning the embedding models; hence alternative solutions are currently investigated [2]. Even more so, existing embedding models do not make use of the additional semantic information encoded within KGs, when more expressive representations are adopted, indeed the need for *semantic embedding methods* has been argued [5, 23, 13].

In this paper we present an approach to graph embeddings that, beyond the graph structural information and properties, is also able to exploit the available knowledge expressed in rich representations like RDFS and OWL. Recent works [19] have proven the effectiveness of combinations of embedding methods and strategies relying on reasoning services for the injection of *Background Knowledge* (BK) to enhance the performance of a specific predictive model. Following this line, we propose TRANSOWL, aiming at injecting BK particularly during the learning process, and its upgraded version TRANSROWL, where a newly defined and more suitable loss function and scoring function are also exploited. Particularly, we focus on the application of this idea to enhance well-known basic scalable models, namely TRANSE [3] and TRANSR [16], the latter tackling some weak points in TRANSE, such as the difficulty of modeling specific types of relationships [2]. We built upon such models to better cope with additional various types of relationships, intervening also on the training process. Indeed, the proposed solutions can take advantage of an informed corruption process that leverages on reasoning capabilities, while limiting the amount of false negatives that a less informed random corruption process may cause.

It is important to note that, in principle, the proposed approach could be applied to more complex (and recent) KG embedding methods. In this work we intended to show the feasibility of the approach, starting with well established models before moving on towards more sophisticated ones, which would need an additional formalization.

The proposed solutions are actually able to improve their effectiveness compared to the original models that focus on structural graph properties with a random corruption process. This is proven through an experimentation focusing on link prediction and triple classification tasks on standard datasets.

The rest of the paper is organized as follows. Basics on KG embedding models that are functional to our method definition are presented in §2. The formalization of our proposed solutions is illustrated in §3 while in §4 the experimental evaluation is provided. Related work is discussed in §5. Conclusions are delineated in §6.

## 2   Basics on Embedding Methods

In the following we assume the reader has familiarity with the standard representation and reasoning frameworks such as RDF, RDFS and OWL, hence we will consider RDF graphs made up of triples $\langle s, p, o \rangle$ of *RDF terms*, respectively the *subject*, the *predicate*, and the *object*, such that $s \in U \cup B$ where $U$ stands for a set of URIs and $B$ as for a set of blank nodes, $p \in U$ and $o \in U \cup B \cup L$ where $L$ stands for a set of *literals*. In the following, given an RDF graph $G$, we denote as $\mathcal{E}_G$ the set of all entities occurring as subjects or objects in $G$, and as $\mathcal{R}_G$ the set of all predicates occurring in $G$.

In this section, basics on *knowledge graph embeddings* methods [4] are recalled, with a special focus on TRANSE [3] and TRANSR [16]. Several models have been actually proposed for embedding KGs in low-dimensional vector spaces, by learning a unique *distributed representation* (or *embedding*) for each entity and predicate in the KG [4] and different representation spaces have been considered (e.g. point-wise, complex, discrete, Gaussian, manifold). Here we focus on vector embedding in the set of real numbers. Regardless of the learning procedure, these models share a fundamental characteristic: given a KG $G$, they represent each entity $x \in \mathcal{E}_G$ by means of a continuous *embedding vector* $\mathbf{e}_x \in \mathbb{R}^k$, where $k \in \mathbb{N}$ is a user-defined hyperparameter. Similarly, each predicate $p \in \mathcal{R}_G$ is associated to a *scoring function* $f_p : \mathbb{R}^k \times \mathbb{R}^k \to \mathbb{R}$, also referred to as *energy function* [20]. For each pair of entities $s, o \in \mathcal{E}_G$, the score $f_p(\mathbf{e}_s, \mathbf{e}_o)$ measures the *confidence* that the statement encoded by $\langle s, p, o \rangle$ holds true.

TRANSE introduces a very simple but effective and efficient model: each entity $x \in \mathcal{E}_G$ is represented by an embedding vector $\mathbf{e}_x \in \mathbb{R}^k$, and each predicate $p \in \mathcal{R}_G$ is represented by a (vector) *translation operation* $\mathbf{e}_p \in \mathbb{R}^k$. The score of a triple $\langle s, p, o \rangle$ is given by the similarity (negative $L_1$ or $L_2$ distance) of the translated subject embedding $(\mathbf{e}_s + \mathbf{e}_p)$ to the object embedding $\mathbf{e}_o$:

$$f_p(\mathbf{e}_s, \mathbf{e}_o) = -\|(\mathbf{e}_s + \mathbf{e}_p) - \mathbf{e}_o\|_{\{1,2\}}. \tag{1}$$

In the case of TRANSR, a different score function $f'_p$ is considered, that preliminarily projects $\mathbf{e}_s$ and $\mathbf{e}_o$ to the different $d$-dimensional space of the relational embeddings $\mathbf{e}_p$ through a suitable matrix $\mathbf{M} \in \mathbb{R}^{k \times d}$:

$$f'_p(\mathbf{e}_s, \mathbf{e}_o) = -\|(\mathbf{M}\mathbf{e}_s + \mathbf{e}_p) - \mathbf{M}\mathbf{e}_o\|_{\{1,2\}}. \tag{2}$$

The optimal embedding and translation vectors for predicates are learned jointly. The method relies on a *stochastic optimization process*, that iteratively updates the distributed representations by increasing the score of the triples in $G$, i.e. the observed

triples $\Delta$, while lowering the score of unobserved triples standing as negative examples contained in $\Delta'$. Unobserved triples are randomly generated by means of a *corruption process*, which replaces either the subject or the object of each observed triple with another entity in $G$. Formally, given an observed triple $t \in G$, let $\mathcal{C}_G(t)$ denote the set of all triples derived by corrupting $t$. Then:

$$\Delta' = \bigcup_{\langle s,p,o\rangle \in \Delta} \mathcal{C}_G(\langle s,p,o\rangle) = \bigcup_{\langle s,p,o\rangle \in \Delta} \{\langle \tilde{s},p,o\rangle \mid \tilde{s} \in \mathcal{E}_G\} \cup \{\langle s,p,\tilde{o}\rangle \mid \tilde{o} \in \mathcal{E}_G\}. \quad (3)$$

The embedding of all entities and predicates in the $G$ is learned by minimizing a *margin-based ranking loss*. Formally, let $\theta \in \Theta$ denote a configuration for all entity and predicate embeddings (i.e. the *model parameters*), where $\Theta$ denotes the parameters space. The optimal model parameters $\hat{\theta} \in \Theta$ is learned by solving a constrained optimization problem that amounts to minimizing the following loss functional:

$$\underset{\theta \in \Theta}{\text{minimize}} \sum_{\substack{\langle s,p,o\rangle \in \Delta \\ \langle \tilde{s},p,\tilde{o}\rangle \in \Delta'}} [\gamma + f_p(\mathbf{e}_s, \mathbf{e}_o) - f_p(\mathbf{e}_{\tilde{s}}, \mathbf{e}_{\tilde{o}})]_+ \quad (4)$$

$$\text{subject to} \quad \forall x \in \mathcal{E}_G \colon \ \|\mathbf{e}_x\| = 1,$$

where $[x]_+ = \max\{0, x\}$, and $\gamma \geq 0$ is a hyperparameter referred to as *margin*.

The loss functional in the problem enforces the score of observed triples to be higher than the score of unobserved triples. The constraints prevent the training process to trivially solve the problem by increasing the entity embedding norms.

## 3   Evolving Models through Background Knowledge Injection

Our approach aims at improving proposed embedding models for KGs, verifying the intuition that the exploitation of expressive schema-level axioms may help increase the model effectiveness. We first present TRANSOWL, injecting BK during the learning process when applied to entity-based models like TRANSE. Then we move on towards a new formalization that is TRANSROWL, which exploits TRANSR, to better handle the various types of relations, besides of adopting a newly defined and more suitable loss function and scoring function.

### 3.1   TRANSOWL

TRANSOWL aims at enhancing simple but effective and efficient entity-based embedding models like TRANSE with a better use of the available BK. The final goal is showing the feasibility of our approach, that in principle can be applied to more complex models with additional formalization. In TRANSOWL the original TRANSE setting is maintained while resorting to reasoning with schema axioms to derive further triples to be considered for training and that are generated consistently with the semantics of the properties. Specifically, TRANSOWL defines specific constraints on the energy functions for each considered axiom, that guide the way embedding vectors are learned. It extends the approach in [19], generating a model characterized by two main components devoted to inject BK in the embedding-based model during the training phase:

**Reasoning:** It is used for generating corrupted triples that can certainly represent negative instances, thus avoiding false negatives, for a more effective model training. Specifically, using a reasoner[3] it is possible to generate corrupted triples exploiting the available axioms, specified in RDFS and OWL, namely domain, range, disjointWith, functionalProperty; moreover, false positives can be detected and avoided.

**BK Injection:** A set of different axioms w.r.t. those mentioned above are employed for the definition of constraints on the energy function considered in the training phase so that the resulting vectors related to such axioms reflect specific properties: equivalentClass, equivalentProperty, inverseOf and subClassOf.

In TRANSOWL the basic loss function minimized in TRANSE (see Eq. 4) is more complex adding a number of terms consistently with the constraints on the energy function based on the underlying axioms. The most interesting one amounts to generating, new triples to be added to the training set on the grounds of the specified axioms. The definition of the loss function along this setting is given as follows:

$$
L = \sum_{\substack{\langle h,r,t \rangle \in \Delta \\ \langle h',r,t' \rangle \in \Delta'}} [\gamma + f_r(h,t) - f_r(h',t')]_+ + \sum_{\substack{\langle t,q,h \rangle \in \Delta_{\text{inverseOf}} \\ \langle t',q,h' \rangle \in \Delta'_{\text{inverseOf}}}} [\gamma + f_q(t,h) - f_q(t',h')]_+
$$

$$
+ \sum_{\substack{\langle h,s,t \rangle \in \Delta_{\text{equivProperty}} \\ \langle h',s,t' \rangle \in \Delta'_{\text{equivProperty}}}} [\gamma + f_s(h,t) - f_s(h',t')]_+ + \sum_{\substack{\langle h,\text{typeOf},l \rangle \Delta \cup \in \Delta_{\text{equivClass}} \\ \langle h',\text{typeOf},l' \rangle \in \Delta' \cup \Delta'_{\text{equivClass}}}} [\gamma + f_{\text{typeOf}}(h,l) - f_{\text{typeOf}}(h',l')]_+
$$

$$
+ \sum_{\substack{\langle h,\text{subClassOf},p \rangle \in \Delta_{\text{subClass}} \\ \langle h',\text{subClassOf},p' \rangle \in \Delta'_{\text{subClass}}}} [(\gamma - \beta) + f(h,p) - f(h',p')]_+ \tag{5}
$$

where $q \equiv r^-$, $s \equiv r$ (properties), $l \equiv t$ and $t \sqsubseteq p$, the sets of triples denoted by $\Delta_\pi$, where $\pi \in \{\text{inverseOf}, \text{equivProperty}, \text{equivClass}, \text{subClass}\}$, represent the additional triples generated by a reasoner exploiting such properties and $f(h,p) = \|\mathbf{e}_h - \mathbf{e}_p\|$. The different formulation for the case of subClassOf is motivated by the fact that it encodes the additional constraint (expressing major specificity) $f_{\text{typeOf}}(e,p) > f_{\text{typeOf}}(e,h)$ where $e$ is an instance, $h$ subClassOf $p$ and $f_{\text{typeOf}}(e,p) = \|\mathbf{e}_e + \mathbf{e}_{\text{typeOf}} - \mathbf{e}_p\|$ as for the original formulation. This also motivates the adoption of the $\beta$ factor, that is required to determine the direction of the inequality to be obtained for the energy values associated to subclass entities (one w.r.t. the other). As for the equivalentClass formulation, the rationale is to exploit as much as possible the information that can be made explicit during the training phase. Particularly, we ground on the fact that the typeOf relation is one of the most common predicates in KGs, and as such it is used as a primary target of the training phase. In order to clarify this aspect, let us consider a class $A$ equivalent to a class $B$. Given the triple $\langle h, \text{typeOf}, A \rangle$ it is possible to derive also $\langle h, \text{typeOf}, B \rangle$. Training the model on those derived triples brings a considerable number of new triples.

### 3.2 TRANSROWL

The main motivation for TRANSOWL was to set up a framework that was able to take into account the BK while using a simple model. However, some of the limits of the

---

[3] Facilities available in the Apache Jena framework were used: `https://jena.apache.org`

models grounded on TRANSE originate from an inability to suitably represent the specificity of the various types of properties. Specifically, the main limitations of TRANSE are related to the poor modeling of reflexive and non 1-to-1 relations as well as to their interplay. Such limitations can cause generating spurious embedding vectors with null values or analogous vectors among different entities, thus compromising the ability of making correct predictions. A noteworthy case regards the typeOf property, a common $N$-to-$N$ relationship. Modeling such property with TRANSE amounts to a simple vector translation; the considered individuals and classes may be quite different in terms of properties and attributes they are involved in, thus determining strong semantic differences (according to [27]) taking place at large reciprocal distances in the underlying vector space, hence revealing the weakness of employing the mere translation.

TRANSR is more suitable to handle such specificity. Thus, further evolving the approach used to derive TRANSOWL from TRANSE, a similar setting is applied to TRANSR, resulting in another model dubbed TRANSROWL, with the variant TRANSROWL$^R$. Particularly, to be more effective on cases involving complex types of relations, a more elaborate vectorial representation is adopted, usually resulting from hyperplane projection [26, 25] or different vector spaces [14, 16]. The underlying model influences the computation of the loss function gradient, that is required to train the model. Hence the main variation introduced by the new model regards the way the entities, within the energy function, are projected in the vector space of the relations, which increases the complexity without compromising the overall scalability. The limitations of TRANSE w.r.t. typeOf can be nearly overcome once the new setting based on TRANSR is adopted. The latter, indeed, associates to typeOf, and to all other properties, a specific vector space where entity vectors are projected to. This leads to training specific projection matrices for typeOf so that the projected entities can be located more suitably to be linked by the vector translation associated to typeOf. Furthermore, methods based on the regularization of the embeddings by exploiting the available axioms [19] prove that resorting to available BK may enhance their effectiveness on account of the addition of more specific constraints to the loss function. As such, the resulting TRANSROWL model maintains the same setting for reasoning adopted by TRANSOWL, but adopt a different utilization of the available axioms in two variants, one analogous to TRANSOWL and the other, dubbed TRANSROWL$^R$, following the method based on the regularization of the embeddings via equivalence and inverse property axioms [19].

TRANSROWL adapts to the TRANSR by introducing in the loss function (Eq. 5) the TRANSR score function $f'(\cdot)$ (Eq. 2) and additional weighting parameters:

$$
\begin{aligned}
L = &\sum_{\substack{\langle h,r,t\rangle \in \Delta \\ \langle h',r,t'\rangle \in \Delta'}} [\gamma + f'_r(h,t) - f'_r(h',t')]_+ + \lambda_1 \sum_{\substack{\langle t,q,h\rangle \in \Delta_{\text{inverseOf}} \\ \langle t',q,h'\rangle \in \Delta'_{\text{inverseOf}}}} [\gamma + f'_q(t,h) - f'_q(t',h')]_+ \\
&+ \lambda_2 \sum_{\substack{\langle h,s,t\rangle \in \Delta_{\text{equivProperty}} \\ \langle h',s,t'\rangle \in \Delta'_{\text{equivProperty}}}} [\gamma + f'_s(h,t) - f'_s(h',t')]_+ + \lambda_3 \sum_{\substack{\langle h,\text{typeOf},l\rangle \in \Delta \cup \Delta_{\text{equivClass}} \\ \langle h',\text{typeOf},l'\rangle \in \Delta' \cup \Delta'_{\text{equivClass}}}} [\gamma + f'_{\text{typeOf}}(h,l) - f'_{\text{typeOf}}(h',l')]_+ \\
&+ \lambda_4 \sum_{\substack{\langle t,\text{subClassOf},p\rangle \in \Delta_{\text{subClass}} \\ \langle t',\text{subClassOf},p'\rangle \in \Delta'_{\text{subClass}}}} [(\gamma - \beta) + f'(t,p) - f'(t',p')]_+
\end{aligned}
\tag{6}
$$

where $q \equiv r^-$, $s \equiv r$ (properties), $l \equiv t$ and $t \sqsubseteq p$ (classes), the different triple sets, denoted by $\Delta_\pi$ with $\pi \in \{\mathsf{inverseOf}, \mathsf{equivProperty}, \mathsf{equivClass}, \mathsf{subClass}\}$, contain additional triples generated by a reasoner exploiting these properties and $f'$ (case of subClass) is defined as for TRANSOWL, considering the embedding vectors coming from TRANSR. The parameters $\lambda_i$, $i \in \{1, \dots, 4\}$, weigh the influence that each function term in Eq. 6 has during the learning phase, analogously to the approach in [19].

In the embedding methods exploiting *axiom-based regularization* [19], the constraints on vectors to be satisfied, representing the related properties of the entities and relations, are explicitly expressed within the loss function. Considering the model TRANSE$^R$ [19], the regularization term based on the equivalence and inverse property axioms is defined as follows:

$$L = \sum_{\substack{\langle h,r,t \rangle \in \Delta \\ (h',r',t') \in \Delta'}} [\gamma + f_r(h,t) - f_r(h',t')]_+ + \lambda \sum_{r \equiv q^- \in \mathcal{T}_{\mathsf{inverseOf}}} \|r + q\| + \lambda \sum_{r \equiv p \in \mathcal{T}_{\mathsf{equivProp}}} \|r - p\| \quad (7)$$

with the hyperparameter $\lambda$ and where $\mathcal{T}_{\mathsf{inverseOf}} = \{r_1 \equiv q_1^-, r_2 \equiv q_2^-, ..., r_n \equiv q_n^-\}$ and $\mathcal{T}_{\mathsf{equivProp}} = \{r_1 \equiv p_1, r_2 \equiv p_2, ..., r_n \equiv p_n\}$ stand for the set of inverse properties and equivalent properties following from the axioms in the BK.

To adapt this approach to TRANSROWL, it is required to include constraints on the considered additional properties, such as equivalentClass and subClassOf, and further constraints on the projection matrices associated to each relation. This variant of TRANSROWL, that is dubbed TRANSROWL$^R$, is formalized as follows:

$$L = \sum_{\substack{\langle h,r,t \rangle \in \Delta \\ \langle h',r',t' \rangle \in \Delta'}} [\gamma + f'_r(h,t) - f'_r(h',t')]_+$$

$$+ \lambda_1 \sum_{r \equiv q^- \in \mathcal{T}_{\mathsf{inverseOf}}} \|r + q\| + \lambda_2 \sum_{r \equiv q^- \in \mathcal{T}_{\mathsf{inverseOf}}} \|M_r - M_q\|$$

$$+ \lambda_3 \sum_{r \equiv p \in \mathcal{T}_{\mathsf{equivProp}}} \|r - p\| + \lambda_4 \sum_{r \equiv p \in \mathcal{T}_{\mathsf{equivProp}}} \|M_r - M_p\|$$

$$+ \lambda_5 \sum_{e' \equiv e'' \in \mathcal{T}_{\mathsf{equivClass}}} \|e' - e''\| + \lambda_6 \sum_{s' \subseteq s'' \in \mathcal{T}_{\mathsf{subClass}}} \|1 - \beta - (s' - s'')\| \quad (8)$$

where $\mathcal{T}_{\mathsf{inverseOf}} = \{r_1 \equiv q_1^-, r_2 \equiv q_2^-, ..., r_n \equiv q_n^-\}$, and $\mathcal{T}_{\mathsf{equivProp}} = \{r_1 \equiv p_1, r_2 \equiv p_2, ..., r_n \equiv p_n\}$, resp. the sets of inverse and of equivalent properties, $\mathcal{T}_{\mathsf{equivClass}} = \{e'_1 \equiv e''_1, e'_2 \equiv e''_2, ..., e'_n \equiv e''_n\}$ and $\mathcal{T}_{\mathsf{subClass}} = \{s'_1 \sqsubseteq s''_1, s'_2 \sqsubseteq s''_2, ..., s'_n \sqsubseteq s''_n\}$ resp. the sets of equivalent classes and subclasses. Parameters $\lambda_i$, $i \in \{1, \dots, 6\}$, determine the weights to be assigned to each constraint and $\beta$ has the same role mentioned above. The additional term for projection matrices is required for inverseOf and equivProp triples to favor the equality of their projection matrices. This is for having the same energy associated, via score function, to the triples in their respective sets. Considering for instance $\langle h, r, t \rangle$ and $\langle h, p, t \rangle$, if $r$ equivProp $p$, their energy should be equal.

The idea of taking into account the regularization of the embeddings by means of the axioms, has been experimentally tested (see Sect. 4) in order to assess whether directly imposing such constraints is more advantageous than generating further triples, based on the same constraints, as in the original definition of the TRANSROWL model.

## 4    Experimental Evaluation

In this evaluation we focused on TRANSOWL, TRANSROWL, TRANSROWL$^R$ compared to the original models TRANSE and TRANSR as a baseline. The evaluation aims at assessing the improvement brought by the choices made for defining new models, grounded on the exploitation of BK injection.

Specifically, we tested the performance of the mentioned models and related systems on the task of *Link Prediction*, together with *Type Prediction* (that, given typeOf-triple for a subject, verifies if the model is able to correctly predict a class the individual belongs to). Then we also tested the models on *Triple Classification* problems, i.e. the ability to classify new triples as true or false. Preliminarily, in the following section, the settings of the experiments are described jointly with the references to the adopted datasets and source code.

### 4.1    Experiment Setup

**Datasets.** The models were tested on four datasets drawn from the following KGs, that have been considered in the experimental evaluations of related works [6, 17].

**DBpedia.** It is a well-known KG with data extracted from Wikipedia. Its vocabulary has 320 classes and 1650 properties. The English version[4] describes 4,58M resources, 4,22M of them classified in an ontology. Its dimensions and the presence of 27M RDF links towards 30+ external sources make it one of the principal reference of the Linked Data cloud. We considered two datasets that were extracted to ensure suitable axioms to test the models under evaluation, namely axioms on domain, range, disjointWith, functionalProperty, equivalentClass, equivalentProperty, inverseOf and subClassOf, in the two variants dubbed[5] *DBpedia100K* [6], containing about 100K entities and 321 relations in 600K triples, and *DBpedia15K*[6] [17], containing about 12.8K entities and 278 relations in 180K triples.

**DBPediaYAGO.** YAGO[7] is a KG organizing knowledge coming from different sources such as *WordNet*, *GeoNames* and *Wikipedia*, including 350K+ classes, 10M entities and 120M assertions [22]. It has been exploited to extend and complete *DBpedia15K*, resulting in *DBPediaYAGO* exploiting the many links connecting to DBpedia. *DBPediaYAGO* is characterized by about 290K triples, with 88K entities and 316 relations.

**NELL.** The dataset[8] comes from a knowledge extraction system for eliciting facts from corpora of Web pages. The resulting KG amounts to 2.810K+ assertions regarding 1.186 different relations and categories. We considered a fragment of NELL2RDF-vanilla[9], that does not contain all of the properties that can be exploited by the proposed model variants. The considered dataset is made up of about 150K triples,

---

[4] https://wiki.dbpedia.org/about

[5] https://github.com/iieir-km/ComplEx-NNE_AER/tree/master/datasets/DB100K

[6] https://github.com/nle-ml/mmkb/tree/master/DB15K

[7] https://yago-knowledge.org/

[8] http://rtw.ml.cmu.edu/rtw/

[9] http://nell-ld.telecom-st-etienne.fr/

with 272 properties and 68K entities. The aim was to test the models on a dataset with a limited set of exploitable properties, namely subClassOf, inverseOf, functionalProperty, disjointWith, range and domain. The abundance of subClassOf-triples, together with a limited number of typeOf-triples for each entity, is meant to test if the models are able to compensate this partial incompleteness and improve the performance of the base models. Considering the inverseOf-axioms allows to compare directly the performance of TRANSROWL and TRANSROWL$^R$, when generating new triples for training or regularizing the embeddings.

**Parameter Settings.** All models were set up along the same procedure and parameter values, consistently with the experiments illustrated in [16, 3]: learning rate: 0.001; minibatch dimension: 50; entity/relation vector dimension = 100; epochs: $\{250, 500, 1000\}$. This choice is motivated by the fact that our first aim is to verify the possible improvements of the proposed solutions over the basic models when exactly the same conditions, including the parameter values, apply.

Due to a tendency to *overfitting* that is known to affect TRANSR, it requires an initialization of the embeddings performed via TRANSE (see [16]). Similarly, also TRANSROWL and its variant were initialized with these embeddings. Overfitting has been checked on the models derived from TRANSR along the different numbers of epochs. Moreover, the `bern` strategy for triple corruption phase was adopted, as this choice led to a better performance compared to the `unif` strategy in previous experimental evaluations of this class of models [26, 16, 14, 25]. The `unif` strategy generates negative triples by sampling a pair of entities for subject and object from $\mathcal{E}_G$, assigning uniform probabilities to the possible replacements; `bern` assigns different chances (along with a Bernoulli distribution) based on the nature (1-to-1,1-to-N, N-to-N) of the relation.

As for the TRANSROWL loss function regularization hyperparameters $\lambda_i$, the following values have been found: inverseOf $\lambda_1 = 1$; equivalentProperty $\lambda_2 = 1$; equivalentClass $\lambda_3 = 0.1$; subClassOf $\lambda_4 = 0.01$; whilst as for TRANSROWL$^R$: $\lambda_1 = \lambda_2 = \lambda_3 = \lambda_4 = \lambda_5 = \lambda_6 = 0.1$;

Each dataset was partitioned into *training*, *validation* and *test* sets by randomly selecting 70%, 10%, 20% of the triples per run. Datasets and their partitions, resulting embedding models, together with the source code are available in a public repository[10].

### 4.2 Link Prediction

Following the standard methodologies for evaluating *Translational Distance Models*, we focus on predicting the missing individuals in given incomplete triples. Specifically the models are used to predict triples $\langle h, r, t \rangle$, with $h, t \in \mathcal{E}_G$ and $r \in \mathcal{R}_G$, corresponding to the patterns $\langle ?, r, t \rangle, \langle h, r, ? \rangle$.

The typical metrics considered for this task are *Mean Rank* and *H@10* (the lower the better, and vice-versa, resp.), that are based on predictions rankings. Two variants are generally taken into account, *Raw* and *Filtered*, where the latter filters off triples that amount to corrupted ones, i.e. negative cases generated for training the model. For

---

[10] `https://github.com/Keehl-Mihael/TransROWL-HRS`

a deeper insight, we measured separately the performance considering all properties but typeOf, and then on typeOf only. This allows to verify the improvement brought by the new models considering *Type Prediction* (i.e. classification) problems on the classes of each KG.

As mentioned above, the embeddings were initialized by a first run of TRANSE (1000 epochs), and the training was run for up to further 1000 epochs. To appreciate the performance trends, for TRANSR and its extensions, we also report the test results for models trained in intermediate numbers of epochs, namely 250 and 500. This is in order to check the occurrence of overfitting cases as discussed above.

The complete outcomes of the link prediction experiments are illustrated in Tab. 1 (best results are bolded, with ties decided by the precise figures).

Preliminarily, comparing the overall performance of TRANSE and TRANSOWL, the latter seems to be able to improve only on classification tasks (those targeting typeOf) and in the experiments on *DBpediaYAGO* and *NELL*, it proves even better, in terms of MR, than TRANSR and derived models. This suggests that TRANSOWL is particularly suitable for classification. However, in most of the cases the best performance on this task was achieved by TRANSROWL especially in terms of *H@10*. Compared to the results achieved by TRANSR, one cannot conclude that the subClassOf axioms have determined the same improvements of TRANSOWL compared to TRANSE, suggesting that more complex models may require more advanced strategies.

Conversely, the results regarding the other properties (no typeOf columns) confirmed that TRANSR and derived models are more suitable for general link prediction problems: TRANSROWL and TRANSROWL$^R$ in most of the cases performed much better than TRANSE and TRANSOWL. Compared to TRANSR, TRANSROWL and TRANSROWL$^R$ showed a better performance, except few cases in which TRANSR resulted slightly better especially in terms of MR, but they were close runner-ups. The improvement w.r.t. TRANSE and TRANSOWL is due to the more suitable representation for the relations. This is more evident from the outcomes on *DBpedia100K* and *DBpediaYAGO*, the latter having been specifically extended to improve the completeness. As argued in [11], a more complete dataset yields a larger number of triples describing single entities/relations, as the resulting prediction model, with more parameters to be fitted, can be better trained.

The case of the *NELL* dataset is more peculiar, as it aimed at testing the models in a condition of larger incompleteness and with a smaller number of properties to be exploited for knowledge injection. Specifically, this dataset is characterized by a much lower number of typeOf-triples per entity, thus making classification a much harder task. This lack is (partly) compensated by a wealth of subClassOf axioms that can be exploited during the training of class vectors (an ability, introduced in TRANSOWL, that is shared also by TRANSROWL and TRANSROWL$^R$). Another type of axioms that abound in the *NELL* dataset is inverseOf. The link prediction results (no typeOf-triples) show a lower performance of both TRANSOWL and TRANSROWL, which suggests that the underlying approach has margins for improvements in its definition and/or calls for a better fitting of the regularization parameters.

Considering the outcomes on intermediate models (after 250 or 500 epochs elapsed) we observe that the methods were not able to improve the resulting models along with

**Table 1.** Link Prediction outcomes (MR = Mean Rank and H@10 = Hits@10)

### DBpedia15K

| model | epochs | no typeOf MR (raw) | H@10 (raw) | MR (flt.) | H@10 (flt.) | typeOf MR (raw) | H@10 (raw) | MR (flt.) | H@10 (flt.) |
|---|---|---|---|---|---|---|---|---|---|
| TRANSE | 1000 | 587.07 | 32.46 | 573.94 | 35.01 | 692.29 | 9.75 | 67.05 | 15.68 |
| TRANSOWL | 1000 | 621.06 | 32.24 | 607.91 | 34.85 | 493.46 | 13.20 | 29.14 | 20.85 |
| TRANSR | 250 | **583.72** | 60.57 | **570.54** | 63.37 | 498.58 | 84.86 | 26.42 | 93.09 |
| TRANSR | 500 | 587.37 | 60.66 | 574.12 | 63.42 | 499.39 | 85.01 | 20.15 | 94.51 |
| TRANSR | 1000 | 600.12 | 60.67 | 586.83 | 63.57 | 504.13 | 85.01 | 13.96 | 95.50 |
| TRANSROWL | 250 | 584.94 | **60.88** | 571.74 | 63.48 | 493.24 | 84.91 | 25.10 | 93.72 |
| TRANSROWL | 500 | 598.03 | 60.77 | 584.79 | 63.58 | 487.44 | 84.97 | 17.50 | 95.38 |
| TRANSROWL | 1000 | 606.73 | 60.59 | 593.45 | 63.48 | **484.04** | **85.18** | **13.53** | **96.54** |
| TRANSROWL$^R$ | 250 | 585.84 | 60.68 | 572.62 | 63.40 | 498.50 | 84.85 | 26.60 | 93.10 |
| TRANSROWL$^R$ | 500 | 592.78 | 60.66 | 579.55 | 63.42 | 491.98 | 84.97 | 19.73 | 95.52 |
| TRANSROWL$^R$ | 1000 | 607.43 | 60.71 | 594.13 | **63.65** | 497.40 | 85.12 | 16.50 | 96.24 |

### DBpedia100K

| model | epochs | no typeOf MR (raw) | H@10 (raw) | MR (flt.) | H@10 (flt.) | typeOf MR (raw) | H@10 (raw) | MR (flt.) | H@10 (flt.) |
|---|---|---|---|---|---|---|---|---|---|
| TRANSE | 1000 | 2233.40 | 38.56 | 2204.39 | 41.11 | 2224.26 | 3.62 | 1615.68 | 3.86 |
| TRANSOWL | 1000 | 2430.51 | 38.12 | 2401.67 | 40.69 | 2152.89 | 5.64 | 1728.52 | 6.02 |
| TRANSR | 250 | 2160.79 | 52.83 | 2131.51 | 55.45 | 1911.06 | 92.21 | 1480.79 | 92.23 |
| TRANSR | 500 | 2152.40 | 53.02 | 2122.94 | 55.67 | 1927.16 | 92.17 | **1479.44** | 92.35 |
| TRANSR | 1000 | 2142.10 | 53.17 | 2112.42 | 55.96 | 1957.42 | 92.04 | 1480.26 | 92.25 |
| TRANSROWL | 250 | 2165.42 | 52.67 | 2136.25 | 55.26 | **1904.80** | 92.22 | 1483.62 | 92.23 |
| TRANSROWL | 500 | 2147.47 | 52.92 | 2118.12 | 55.59 | 1933.79 | 92.22 | 1498.14 | 92.37 |
| TRANSROWL | 1000 | 2147.56 | **53.24** | 2117.87 | **56.03** | 1961.75 | **92.29** | 1503.87 | **92.43** |
| TRANSROWL$^R$ | 250 | 2159.51 | 52.76 | 2130.29 | 55.35 | 1915.67 | 91.98 | 1485.03 | 92.18 |
| TRANSROWL$^R$ | 500 | 2136.73 | 52.92 | 2107.29 | 55.64 | 1955.90 | 92.07 | 1515.07 | 92.27 |
| TRANSROWL$^R$ | 1000 | **2121.52** | 53.08 | **2091.81** | 55.95 | 1971.98 | 92.24 | 1511.07 | **92.43** |

### DBpediaYAGO

| model | epochs | no typeOf MR (raw) | H@10 (raw) | MR (flt.) | H@10 (flt.) | typeOf MR (raw) | H@10 (raw) | MR (flt.) | H@10 (flt.) |
|---|---|---|---|---|---|---|---|---|---|
| TRANSE | 1000 | 7417.08 | 19.24 | 7385.12 | 20.20 | 587.19 | 8.71 | **157.14** | 19.42 |
| TRANSOWL | 1000 | 7455.49 | 19.21 | 7423.56 | 20.18 | **580.29** | 8.68 | 162.03 | 19.43 |
| TRANSR | 250 | 7279.11 | 44.04 | 7247.16 | 45.13 | 656.10 | 83.66 | 187.91 | 93.47 |
| TRANSR | 500 | 7256.86 | 44.03 | 7224.74 | 45.18 | 738.33 | 81.33 | 249.10 | 88.68 |
| TRANSR | 1000 | 7271.50 | **44.64** | 7239.09 | **46.07** | 844.51 | 81.98 | 348.65 | 88.99 |
| TRANSROWL | 250 | 7279.37 | 43.76 | 7247.37 | 44.92 | 702.22 | **84.48** | 243.45 | **94.54** |
| TRANSROWL | 500 | 7274.77 | 43.94 | 7242.67 | 45.13 | 796.46 | 83.44 | 314.03 | 92.93 |
| TRANSROWL | 1000 | 7209.02 | 44.45 | 7176.64 | 45.84 | 868.27 | 82.81 | 373.90 | 91.17 |
| TRANSROWL$^R$ | 250 | 7274.52 | 43.61 | 7242.52 | 44.78 | 667.70 | 83.21 | 208.90 | 93.22 |
| TRANSROWL$^R$ | 500 | **7196.12** | 44.15 | **7164.00** | 45.34 | 752.57 | 82.08 | 271.03 | 90.52 |
| TRANSROWL$^R$ | 1000 | 7226.55 | 44.13 | 7194.21 | 45.52 | 845.42 | 81.71 | 352.16 | 88.77 |

### NELL

| model | epochs | no typeOf MR (raw) | H@10 (raw) | MR (flt.) | H@10 (flt.) | typeOf MR (raw) | H@10 (raw) | MR (flt.) | H@10 (flt.) |
|---|---|---|---|---|---|---|---|---|---|
| TRANSE | 1000 | 7162.08 | 19.01 | 6969.07 | 26.54 | 2872.45 | 6.55 | 2708.90 | 6.82 |
| TRANSOWL | 1000 | 9622.40 | 15.54 | 9423.73 | 21.72 | **2263.09** | 6.52 | **2092.51** | 6.92 |
| TRANSR | 250 | 7118.13 | 47.13 | 6921.77 | 55.10 | 2796.70 | 79.28 | 2628.58 | 79.70 |
| TRANSR | 500 | 6928.74 | 47.31 | 6728.67 | 55.62 | 2585.97 | 79.19 | 2415.26 | 79.66 |
| TRANSR | 1000 | **6891.20** | **47.40** | **6681.76** | **55.93** | 2315.08 | 79.94 | 2140.16 | 80.50 |
| TRANSROWL | 250 | 7263.08 | 46.76 | 7066.55 | 54.72 | 2775.22 | 79.05 | 2606.66 | 79.40 |
| TRANSROWL | 500 | 7005.75 | 46.86 | 6804.32 | 55.07 | 2545.47 | 79.44 | 2374.09 | 79.86 |
| TRANSROWL | 1000 | 7136.77 | 46.72 | 6929.10 | 55.40 | 2334.50 | **80.00** | 2161.67 | **80.56** |
| TRANSROWL$^R$ | 250 | 7530.80 | 45.89 | 7334.86 | 53.51 | 2714.23 | 78.40 | 2547.79 | 78.75 |
| TRANSROWL$^R$ | 500 | 7300.14 | 46.04 | 7098.93 | 53.89 | 2527.41 | 79.81 | 2357.55 | 80.20 |
| TRANSROWL$^R$ | 1000 | 7339.53 | 46.09 | 7132.22 | 54.15 | 2310.11 | 79.52 | 2138.99 | 80.21 |

the iterations: in a few cases the overall best results were achieved by models trained after fewer epochs had elapsed. This suggests that a more refined regularization would be required.

Lastly, we noticed that TRANSROWL and TRANSROWL$^R$ turned out substantially equivalent in terms of effectiveness, thus indicating efficiency as a criterion for the choice between the alternatives.

### 4.3  Triple Classification

*Triple Classification* is another *KG refinement* task that focuses on discerning correct from incorrect triples with respect to the KG. Also for this task, the way for evaluating predictive models has be consistent with the KG embedding methods.

The evaluation procedure introduced in [24] measures the ability to predict whether a triple is positive or negative, i.e. it represents a true or false fact w.r.t. the KG. To make this decision, a threshold $s_r$ is to be determined for each $r \in \mathcal{R}_G$ so to maximize the *False Positive Rate* (FPR), then test triples will be deemed as positive when their energy-based score is greater than $s_r$, and negative otherwise [18, 26]. The value for $s_r$ was estimated considering a random sample of $r$-triples selected from the training set. They represent the triples that the model has learned to deem as true; for each sampled triple the energy value is computed, measuring the degree of likelihood associated to the triple, setting the threshold $s_r$ to the minimum value. The ability of the model to correctly classify triples is evaluated considering the thresholds obtained for the single relations; this unavoidably increases the chance of predicting as true triples that are actually false, thus it allows to better evaluate the model robustness on the classification of typeOf-triples (especially with simple models such as TRANSE).

Analogously to the previous experiments the performance indices were determined separating the cases of typeOf-triples from those involving the other properties. This allows to better focus on the performance of the proposed models on this relation. The corrupted (negative) triples required for the tests, were generated by reasoning on range and domain axioms for the experiment excluding typeOf, while disjointWith axioms were exploited to get false typeOf-triples.

The experimental setting is analogous to the first part (see §4.1). Tab. 2 reports the complete results for each dataset in terms of accuracy, precision, recall, and false positive rate. Focusing preliminarily on the results of TRANSE and TRANSROWL, we can appreciate a general improvement of the latter especially in terms of FPR (typeOf problems) and in terms of accuracy and recall in (no typeOf) experiments on two datasets in which it outperformed also the other models. The overall results show that TRANSROWL and TRANSROWL$^R$, achieve the best performance, with a few exceptions, particularly in terms of FPR, also on account of a higher precision and limited decays in terms of recall thus resulting in comparable accuracy measures. These similar performance are likely due to the similar formulation of the respective loss function. In the case of TRANSROWL it determines the generation of further triples, based on the specified axioms, used to train the models: all entities and relations are involved in training. Conversely, in the case of TRANSROWL$^R$, only entities and relations that comply with the properties in the constraints are considered. This may explain the slightly superior performance of TRANSROWL. Analogously to the experiments in §4.2, a more

**Table 2.** Triple Classification outcomes (Accuracy, Precision, Recall and FP Rate)

| model | epochs | no typeOf Acc. | P | R | FPR | typeOf Acc. | P | R | FPR |
|---|---|---|---|---|---|---|---|---|---|
| **DBpedia15K** | | | | | | | | | |
| TRANSE | 1000 | 0.663 | 0.991 | 0.407 | 0.006 | 0.899 | 0.781 | 0.958 | 0.865 |
| TRANSOWL | 1000 | **0.658** | 0.967 | **0.407** | 0.023 | 0.975 | 0.990 | 0.933 | 0.127 |
| TRANSR | 250 | 0.655 | 0.998 | 0.390 | 0.002 | 0.961 | 0.928 | 0.954 | 0.616 |
| TRANSR | 500 | 0.650 | 0.996 | 0.380 | 0.002 | 0.978 | 0.979 | 0.953 | 0.303 |
| TRANSR | 1000 | 0.641 | 0.998 | 0.364 | 0.001 | 0.972 | 0.966 | 0.946 | 0.378 |
| TRANSROWL | 250 | 0.646 | 0.996 | 0.373 | 0.003 | 0.969 | 0.924 | **0.987** | 0.857 |
| TRANSROWL | 500 | 0.652 | 0.997 | 0.385 | 0.002 | **0.985** | 0.993 | 0.960 | 0.141 |
| TRANSROWL | 1000 | 0.631 | 0.997 | 0.347 | 0.002 | 0.962 | **0.999** | 0.882 | **0.006** |
| TRANSROWL$^R$ | 250 | 0.648 | 0.997 | 0.377 | 0.002 | 0.937 | 0.989 | 0.816 | 0.049 |
| TRANSROWL$^R$ | 500 | 0.647 | 0.997 | 0.376 | 0.002 | 0.938 | 0.994 | 0.815 | 0.027 |
| TRANSROWL$^R$ | 1000 | 0.628 | **0.998** | 0.342 | **0.001** | 0.981 | 0.969 | 0.972 | 0.523 |
| **DBpedia100K** | | | | | | | | | |
| TRANSE | 1000 | 0.742 | 0.993 | 0.390 | 0.004 | 0.958 | 0.667 | **0.943** | 0.891 |
| TRANSOWL | 1000 | 0.714 | 0.901 | 0.359 | 0.058 | 0.980 | 0.908 | 0.835 | 0.337 |
| TRANSR | 250 | 0.730 | 0.997 | 0.359 | 0.001 | 0.983 | 0.890 | 0.900 | 0.526 |
| TRANSR | 500 | 0.721 | 0.998 | 0.337 | 0.001 | 0.980 | 0.853 | 0.910 | 0.635 |
| TRANSR | 1000 | 0.711 | 0.998 | 0.313 | 0.001 | 0.976 | 0.884 | 0.800 | 0.344 |
| TRANSROWL | 250 | **0.744** | 0.998 | **0.392** | 0.001 | 0.983 | 0.924 | 0.851 | 0.321 |
| TRANSROWL | 500 | 0.730 | 0.995 | 0.361 | 0.003 | 0.979 | **0.965** | 0.768 | 0.106 |
| TRANSROWL | 1000 | 0.705 | 0.998 | 0.300 | **0.001** | **0.987** | 0.940 | 0.895 | 0.353 |
| TRANSROWL$^R$ | 250 | 0.732 | 0.997 | 0.364 | 0.002 | 0.952 | 0.635 | 0.936 | 0.893 |
| TRANSROWL$^R$ | 500 | 0.717 | 0.997 | 0.328 | 0.002 | 0.971 | 0.951 | 0.668 | **0.094** |
| TRANSROWL$^R$ | 1000 | 0.704 | 0.998 | 0.298 | 0.001 | 0.981 | 0.872 | 0.890 | 0.543 |
| **DBpediaYAGO** | | | | | | | | | |
| TRANSE | 1000 | 0.654 | 0.914 | 0.428 | 0.066 | 0.962 | 0.969 | 0.841 | 0.144 |
| TRANSOWL | 1000 | **0.692** | 0.887 | **0.441** | 0.091 | 0.931 | 0.961 | 0.688 | 0.081 |
| TRANSR | 250 | 0.658 | 0.953 | 0.331 | 0.024 | 0.885 | 0.965 | 0.449 | 0.029 |
| TRANSR | 500 | 0.656 | 0.964 | 0.325 | 0.017 | 0.861 | 0.955 | 0.335 | 0.023 |
| TRANSR | 1000 | 0.644 | 0.964 | 0.300 | 0.016 | 0.844 | 0.946 | 0.247 | 0.018 |
| TRANSROWL | 250 | 0.662 | 0.965 | 0.336 | 0.018 | **0.980** | 0.982 | **0.919** | 0.170 |
| TRANSROWL | 500 | 0.658 | 0.964 | 0.328 | 0.018 | 0.867 | **0.988** | 0.351 | **0.006** |
| TRANSROWL | 1000 | 0.649 | 0.968 | 0.307 | 0.014 | 0.905 | 0.973 | 0.547 | 0.032 |
| TRANSROWL$^R$ | 250 | 0.651 | 0.963 | 0.315 | 0.017 | 0.876 | 0.965 | 0.406 | 0.024 |
| TRANSROWL$^R$ | 500 | 0.648 | 0.978 | 0.302 | 0.010 | 0.864 | 0.959 | 0.349 | 0.023 |
| TRANSROWL$^R$ | 1000 | 0.636 | **0.981** | 0.277 | **0.007** | 0.854 | 0.953 | 0.299 | 0.020 |
| **NELL** | | | | | | | | | |
| TRANSE | 1000 | 0.733 | 0.755 | **0.691** | 0.420 | 0.626 | 0.276 | 0.900 | 0.959 |
| TRANSOWL | 1000 | 0.675 | 0.677 | 0.671 | 0.493 | **0.819** | **0.430** | 0.615 | 0.680 |
| TRANSR | 250 | 0.751 | 0.810 | 0.656 | 0.311 | 0.715 | 0.305 | 0.672 | 0.823 |
| TRANSR | 500 | 0.751 | 0.819 | 0.644 | 0.285 | 0.749 | 0.311 | 0.544 | 0.726 |
| TRANSR | 1000 | **0.758** | 0.843 | 0.636 | 0.245 | 0.803 | 0.389 | 0.519 | **0.630** |
| TRANSROWL | 250 | 0.745 | 0.816 | 0.632 | 0.279 | 0.562 | 0.246 | **0.911** | 0.969 |
| TRANSROWL | 500 | 0.744 | 0.815 | 0.633 | 0.282 | 0.735 | 0.311 | 0.610 | 0.776 |
| TRANSROWL | 1000 | 0.744 | 0.835 | 0.608 | 0.234 | 0.763 | 0.334 | 0.560 | 0.717 |
| TRANSROWL$^R$ | 250 | 0.737 | 0.807 | 0.621 | 0.281 | 0.634 | 0.268 | 0.804 | 0.919 |
| TRANSROWL$^R$ | 500 | 0.743 | 0.830 | 0.612 | 0.245 | 0.723 | 0.293 | 0.583 | 0.771 |
| TRANSROWL$^R$ | 1000 | 0.739 | **0.845** | 0.587 | **0.207** | 0.760 | 0.337 | 0.598 | 0.745 |

incomplete dataset like *NELL* turned out to be more difficult for methods relying on a rich BK, whereas a more complete dataset like *DBpediaYAGO*, yielded a better performance of the newly proposed models with differences between the problems focusing on/excluding typeOf-triples.

Considering the outcomes on intermediate models again there is no clear indication of improvement with the elapsing of the epochs on all performance indexes. This may suggest that involving targeted objectives in the training loop may help.

## 5    Related Work

The presented knowledge injection approach could be applied to many other embedding models [4] with the aim of exploiting the rich schema-level axioms often available in the context of the Semantic Web. However, various approaches have been proposed that leverage different specific forms of prior knowledge to learn better representations exploited for KG refinement tasks.

In [9] a novel method was proposed jointly embedding KGs and logical rules, where triples and rules are represented in a unified framework. Triples are represented as atomic formulae while rules are represented as more complex formulae modeled by t-norm fuzzy logics admitting antecedents single atoms or conjunctions of atoms with variables as subjects and objects. A common loss over both representation is defined which is minimized to learn the embeddings. The specific form of BK which has to be available for the KG constitutes the main drawback of these approaches.

In [21] a solution based on adversarial training is proposed that exploits Datalog clauses to encode assumptions which are used to regularize neural link predictors. An inconsistency loss is derived that measures the degree of violation of such assumptions on a set of adversarial examples. Training is defined as a minimax problem, in which the models are trained by jointly minimizing the inconsistency loss on the adversarial examples jointly with a supervised loss. A specific form of BK is required and a specific form of *local CWA* is assumed to reason with it. The availability of such clauses and the assumptions on their semantics represent the main limitations of this approach. Another neural-symbolic approach exploiting prior knowledge through *Logic Tensor Networks* [7] has been applied to similar classification tasks.

A common shortcoming of the related methods is that BK is often not embedded in a principled way. In [10], investigating the compatibility between ontological knowledge and different types of embeddings, they show that popular embedding methods are not capable of modeling even very simple types of rules, hence they are not able to learn the underlying dependencies. A general framework is introduced in which relations are modeled as convex regions which exactly represent ontologies expressed by a specific form of rules, that preserve the semantics of the input ontology.

In [1] the limitations of the current embedding models were identified: theoretical inexpressiveness, lack of support for inference patterns, higher-arity relations, and logical rule incorporation. Thus, they propose the translational embedding model BoxE which embeds entities as points, and relations as a set of hyper-rectangles, which characterize basic logical properties. This model was shown to offer a natural encoding for many logical properties and to be able to inject rules from rich classes of rule languages.

## 6    Conclusions and Ongoing Work

We have proposed an approach to learn embedding models based on exploiting prior knowledge both during the learning process and the triple corruption process to improve the quality of the low-dimensional representation of knowledge graphs. New models have been defined TRANSOWL, TRANSROWL and TRANSROWL$^R$, implemented as publicly available systems. An experimental evaluation on knowledge graph refinement tasks has proved the improvements of the derived models compared to the original ones, but also some shortcomings that may suggest valuable research directions to be pursued.

We are currently working on the application of the presented approach to newer embedding models which have been proved more effective than those considered in this work. We intend to extend the approach by exploiting further schema-axioms as well as hierarchical patterns on properties that can be elicited from the embeddings, namely clusters of relations and hierarchies of sub-relations. We are also planning to apply embedding models for solving other predictive problems related to the KGs. Following some previous works, further methods based on embedding spaces induced by specific kernel functions will also be investigated.

*Acknowledgment:*  We would like to thank Giovanni Sansaro who formalized and developed the code for the preliminary version of TransOWL for his bachelor thesis.

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
