# OpenReview forum: "Injecting Background Knowledge into Embedding Models for Predictive Tasks on Knowledge Graphs"
_eswc-conferences.org/ESWC/2021/Conference/Research_Track — ESWC 2021 Research_

### Official Review · AnonReviewer4 · 2021-01-11
**Minor iterative improvement on previous work that suffers from poor presentation and unconvincing evaluation**

**Confidence:** 5
**Impact:** 2
**Design And Technical Quality:** 2

**Review:**

The paper proposes an adaptation to the established [TransR model](https://www.aaai.org/ocs/index.php/AAAI/AAAI15/paper/view/9571) for knowledge graph embedding in order to use schema axioms ($\mathsf{inverseOf}$, $\mathsf{equivProperty}$, $\mathsf{equivClass}$, and $\mathsf{subClass}$) to improve the learned embeddings: instances of these axioms are used to enrich the training datasets of positive and corrupted triple samples, respectively.
One of the advantages of the presented idea is that theoretically it could not just be applied to TransR, but also to the plethora of more recent knowledge graph embedding methods published in recent years.
However, the paper itself makes no mention of this!

Indeed much of the paper reads like there has been little work on knowledge graph embedding since the TransR model in 2015.
While there are some references to more recent models and survey papers, these are mostly just used to reference datasets or standard tasks.

The first four pages of the paper are spent on a lengthy introduction to standard concepts in the field, i.e., definition of what an embedding model is, introduction of the tasks of link prediction and triple classification, definition of the well-known TransE model and its learning process, etc.
This introduction is fairly well written and should be understandable to newcomers to the field.
But almost all of it well-known in the field and thanks to its popularity also in the wider semantic web community, so that the first two sections should instead have just referred to one of the existing survey papers and then could be condensed considerably.
Conversely, the introduction should have said more about what the presented method actually *is* because this was still unclear to me after reading the first section.
The TransR model, which is the main building block used in the paper, is not defined at all.

The main part of the paper then builds upon references \[22], which introduced the TransOWL model, and \[16], which introduced some regularization terms for TransE.
Compared to the lengthy introduction these two papers are not summarized *at all*.
For both just a single equation with barely any explanation is presented, and one has to read extra carefully that these are indeed not contributions of the paper under review.
Perhaps the authors expected readers to be familiar with these references?
Well, reference \[22] is an Italian bachelor thesis written at the authors' university, which, additionally, I was not able to find on the web.
The models presented in the paper under review, now, only appear to be variants of those from \[16] and \[22], in which the base of TransE has been substituted for TransR.
I could not identify any further contribution.
Besides references \[16] and \[22] no mention of other related work that also enriches training data or uses new regularization terms is made.

The evaluation follows standards of the field and compares link prediction and triple classification performance on a number of datasets.
TransE, TransR, and TransOWL are the only baselines that are compared against.
All evaluated models appear to be very similar in performance, with the best performing model changing depending on number of trained epochs, evaluation tasks, *and* evaluation metric.
To me, no pattern is discernible for in which cases which model performs substantially better than the baselines.
Since each model configuration seems to have been evaluated exactly once, most performance differences are likely not statistically significant.
No mention is made that any hyperparameter optimization was performed.

The authors implementation of the presented models and all datasets seem to be available in the linked GitHub repository and seem fairly well documented.
I am therefore confident, that the presented results are reproducible.

Overall, the paper presents only a minor iterative improvement over previous work that neither shines through a elegant presentation nor a convincing evaluation.
In principle, the presented idea of using axioms to enrich the training samples is promising, but it should have been presented as a possible improvement that is applicable to a wide range of existing knowledge graph embedding methods including more recent ones, e.g., [ComplEx](http://proceedings.mlr.press/v48/trouillon16.html), [ConvE](https://www.aaai.org/ocs/index.php/AAAI/AAAI18/paper/view/17366), or [RotatE](https://openreview.net/forum?id=HkgEQnRqYQ).
I therefore encourage the authors to continuing working on this approach and showing its true potential.

Minor points:

- Section 2 defines that all knowledge graph embeddings represent entities as a vectors $\mathbf{e}_x \in \mathbb{R}^k$ and that all scoring functions take the form $f_p : \mathbb{R}^k \times \mathbb{R}^k \to \mathbb{R}$.
  This is of course a gross simplification, which is not pointed out by the authors, but they themselves say in section 1 that representation in other spaces exist, e.g., complex, Gaussian, etc.

- I did not really understand section 3.1.

- I don't understand all details of the presented model in section 3.2:

    - Eq (1): If we have $\langle h, r, t \rangle \in \Delta$ and $\langle h, \mathsf{typeOf}, l \rangle \in \Delta_\mathsf{equivClass}$ why can we not use these two independently from one another for training, but need to further know that $l \equiv t$?
      (Similar for the other relations.)
    - Eq (2): The difference of just additional coefficients in comparison to Eq (1) could have been communicated more efficiently in the text.
      I am not sure, why these coefficient are needed now and are not used with the TransOWL model.
      I am further assuming that the actual scoring function $f_r$ looks substantially different between both TransOWL (Eq. (1)) and TransROWL (Eq. (2)), but this is not evident from the notation at all.
      I suggest to use something like $f_r^\mathrm{TransE}$ instead.
    - I don't understand why the $\beta$ is specifically needed with $\mathsf{subClassOf}$, and it is not explained anywhere in the text.
    - I don't understand the definitions of $\mathcal{T}_\mathsf{inverseOf}$, $\mathcal{T}_\mathsf{equivProp}$, and $\mathcal{T}_\mathsf{subClass}$, although I can imagine what the idea behind them is.
    - There is no detailed explanation on how the $\Delta_\pi$ and $\Delta'_\pi$ sets are constructed.

- For the training of the TransR, TransROWL, and TransROWL^R models it is pointed out that they are initialized with a TransE embedding trained over 1000 epochs.
  One must conclude that these models were trained for more epochs than the TransE baseline, but more importantly also the TransOWL baseline, making the presented results not really comparable.
  Would it not have been better to compare against TransE and TransOWL models that were trained for 1250, 1500, and 2000 epochs, respectively?
  Additionally, overfitting is mentioned as a reason for why models trained for fewer epochs sometimes perform better.
  To judge this interpretation, a plot which showcases model performance with respect to training epochs (and with a higher resolution than just 250, 500, 1000) would have been desirable.

- The evaluation results are split for data that contains no $\mathsf{typeOf}$ relation and for data that only contains the $\mathsf{typeOf}$ relation.
  I am not convinced that this is not hand-picking the one case where the proposed models performs better.
  In any case, it would have been nice to have additional columns that also present the results on the full dataset, to see how much influence the model would actually have in practice.

- I am not familiar with the triple classification metric FPR and also could not find it in the referenced papers.
  I assume this is just a standard $F_1$ measure? If yes, why call it differently?

- The method of finding the triple classification threshold $s_r$ surprised me.
  Why not take a sample of some positive and corrupted triples from the training set, and then select the threshold that maximizes the metric for that sample?

- During the interpretation of the triple classification results it is suddenly mentioned that specifically this part was trained with a different optimizer than the other configurations.
  However, this basically calls into question large parts of the evaluation.
  Are the improvements seen compared to TransR due actually through the presented method or just through the optimizer?
  In my view, the point of this paper is not to reach the absolute best evaluation metrics (which it does not reach in any case) but rather to study whether existing models can be improved by enriching training data.
  This is not really concludable if also other aspects of the model have been changed which are not related to that idea.

- A distinction of methods "bern" and "unif" is mentioned, but it is not actually explained what this is, nor is a reference given (this is relatively standard though).

- The implementation seems to be rather complicated.
  Out of interest: from my understanding the TransROWL model (i.e. the one without any extra regularization terms) could have also been implemented by just leveraging one of the existing TransR implementations by any of the established knowledge graph embedding frameworks (e.g., [PyKeen](https://github.com/pykeen/pykeen), [AmpliGraphy](https://github.com/Accenture/AmpliGraph), [OpenKE](https://github.com/thunlp/OpenKE), etc.) and substituting modified `train.txt`, `valid.txt`, and `test.txt` with data enriched via the schema axioms?

- For the most part the paper is written in decent, understandable English.
  I stumbled over some infrequent grammar errors.
  Typos I encountered:

    - Section 1: "\[...] as *much* comprehensive as possible \[...]"
    - Section 1: "\[...] method*s* that \[...] *is* able to better exploit \[...]"
    - Section 3.2: "\[...] expressed *withing* the loss function."
    - Section 4.1: "\[...] partitionaning \[...]"
    - Section 4.1: "\[...] per*ff*ormance \[...]"
    - Section 4.2 "\[...] in § 4.1, *And* the \[...]"
    - Section 4.2 "\[...] based on *TransRwere* trained \[...]"
    - References are inconsistently cited, with both "Proceedings" and "Proc." as well as abbreviated journals and unabbreviated journals occurring.

**Anonymity:**

Yes, I would like my review to remain anonymous.

**Rating:**

-2: Reject

**Reuse And Availability:**

4: High

**Strong Points:**

- Straightforward idea for modifying TransR, which, in theory, is also applicable to many other knowledge graph embedding models.
- Publicly available implementation and datasets.

**Subreviewer:**

I submitted this review.

**Weak Points:**

- Minor iterative improvement on a relatively old model, where better performing alternatives would have been available.
- Unconvincing evaluation with unclear results and weak baselines.
- Presentation that talks to much about basics of the knowledge graph embedding field and much to little about approaches it builds upon and related work.

---

> ### Author Rebuttal · Authors · 2021-01-29
>
> “[...] presented idea could not just be applied to TransR, but to the plethora of more recent KG embedding methods” __R:__ The reviewer is right. We must emphasize the orthogonality of the method for exploiting the BK and the model/method it is applied to. In this work we intended to show the feasibility of a solution based on regularization, starting with well established models before moving on towards more sophisticated and current models. The mentioned ultimate goal still requires additional work to come up with a formalization. We will clarify this aspect in the final version of the paper.
>
> “The models presented in the paper [...] I could not identify any further contribution.” __R:__ On the ground of the idea of exploiting expressive schema-level axioms for improving the model effectiveness, we started with proposing TransOWL and its upgraded version TransROWL. Please note that actually TransOWL is part of the paper contribution. We compressed the presentation of TransOWL due to lack of space, preferring to dedicate more space to TransROWL that constitutes a significant improvement of TransOWL. Indeed, as the reviewer may notice, TransOWL has not been published. We reported [22], which is a bachelor thesis, as a form of acknowledgement to the student working on the preliminary version of TransOWL. We have clearly understood that this resulted confusing and reduced the perception of novelty and advances of our contribution. We will make this aspect absolutely clear, in the final version of the paper
>
> “[...] hyperparameter optimization [...]” __R:__ As for the parameter setting, for the sake of comparison, the very same values coming from the original papers have been adopted. We will make this clear in the final version of the paper
>
> “Section 2 defines [...]” __R:__ As for the scoring function, we will specify that other domains may be considered (as already pointed out) but we focus on real valued functions similarly to the targeted methods.
>
> “[...] sect. 3.1” __R:__ This section is meant to recall limitations of the base models and related solutions to specific problems that have been proposed in the literature. It aimed at motivating our BK injection resulting in the proposed model TransOWL further evolved in TransROWL.
>
> “[...] sect. 3.2:”
>
> >“Eq (1): [...]” __R:__ The motivation is straightforward: if we have the triple (h,r,t) and we also know that t is equivalentClass to l than we can derive that (h,r,l) holds and if r= typeOf then (h,typeOf,l) holds too.
>
> >“Eq (2): [...]” __R:__ We agree with the reviewer. Similarly to [16], we introduced additional coefficients to better regulate the contribution of those terms in the training phase. We will make this aspect explicit in the final version of the paper.
>
> >“[...] \Beta [...]” __R:__ \Beta has the purpose to calculate a different value for the margin hyperparameter \gamma; this is due to the different formulation of the score function in the case of the subclassOf, which aims at assigning a lower energy value to the triples that contain subclasses wrt the same triples but that involve the super class. We will add the explanation in the final version of the paper.
>
> >“[...] T_inverseOf [...]” __R:__ they stand for the set of equivalence classes, equivalent properties etc. and follow from the axioms that we consider, as illustrated at the beginning of section 3.2 “BK Injection”. We will specify this after introducing Eq. (1)
>
> >“[...] \Delta_pi and [\Delta’_pi] [...]” __R:__ Delta_pi contains the corresponding triples from the KG. \Delta’_\pi is constructed using the reasoner, “unif” or”bern” as specified in sect. 4. We will make explicit this aspect also in sect. 3.2.
>
> “For the training of the TransR, TransROWL, [...] a plot which showcases [...]” __R:__ We adopted the very same setting as for TransR. Regarding the suggestion of plotting the model performance wrt the training epochs, we will take it into account.
>
> “[...] split for data that contains no typeOf relation and for data that only contains the typeOf relation. [...]” __R:__ The introduction of the model TransR as support for TransROWL is due to the necessity of better representing predicates like typeOf. In order to better evaluate the increase in performance in this aspect, we splitted the evaluation phase considering exclusively the typeOf relation. We also considered the whole dataset without distinctions, but, due to lack of space in the paper, we decided not to report it because the results were comparable.
>
> “[...] FPR” __R:__ It stands for False Positive Rate. We will report the full name version besides the acronym.
>
> “[...] different optimizer [...]” __R:__ we thank the reviewer for highlighting the issue. Any different optimizer has been used. We will correct the text for the revised version.
>
> “[...] "bern" and "unif" [...]” __R:__ Reference [25], was already reported in the paper (actually jointly with other references). We will also add a short explanation of the two methods.

---

> > ### Comment · AnonReviewer4 · 2021-02-03
> > **Response to Rebuttal**
> >
> > Thanks for the elaborate response, and especially for planning to update the paper with regard to my listed minor points. Based on your rebuttal, I want to expand upon two points of my review:
> >
> > > “The models presented in the paper [...] I could not identify any further contribution.” R: On the ground of the idea of exploiting expressive schema-level axioms for improving the model effectiveness, we started with proposing TransOWL and its upgraded version TransROWL. Please note that actually TransOWL is part of the paper contribution. We compressed the presentation of TransOWL due to lack of space, preferring to dedicate more space to TransROWL that constitutes a significant improvement of TransOWL. Indeed, as the reviewer may notice, TransOWL has not been published. We reported [22], which is a bachelor thesis, as a form of acknowledgement to the student working on the preliminary version of TransOWL. We have clearly understood that this resulted confusing and reduced the perception of novelty and advances of our contribution. We will make this aspect absolutely clear, in the final version of the paper
> >
> > While reviewing, I of course had in mind that reference [22] is a bachelor thesis, which being unpublished work you naturally wanted to turn into a paper.
> > For cases where the student does not contribute sufficiently to be co-author of the paper, I like the position of citing the thesis as at least some acknowledgement - in fact, I try to do so in my own papers too.
> > However, in my view the main downside of the submitted paper is that it presents the TransOWL approach almost exclusively as related work, and not with the extended presentation that it would deserve as the core contribution.
> > Which is why I worded my review as I did.
> > Additionally, to me, TransROWL is basically an incremental improvement on top of TransOWL, and should be then condensed as such.
> > In summary, I feel like the main points of my review and its score still stand, and that rewriting the paper on such a core level as I am suggesting here would require another submission.
> >
> > > “[...] hyperparameter optimization [...]” R: As for the parameter setting, for the sake of comparison, the very same values coming from the original papers have been adopted. We will make this clear in the final version of the paper
> >
> > While I understand where you are coming from, I don't feel that this is sufficient justification, for not at least tuning the learning rate (which reviewer 5 also mentioned).
> > Because while your approach is a modification of TransR, this does not imply that it will be optimal under the same hyperparameter configuration.

---

### Official Review · AnonReviewer2 · 2021-01-12
**TransROWL -- injecting the schema/ontology into KG embedding**

**Confidence:** 4
**Impact:** 3
**Design And Technical Quality:** 2

**Review:**

This paper presents a method named TransROWL which improves TransOWL by replacing its TransE style translation to TransR style translation. It further presents a new method TransROWL^R which adds the axiom-based regularization to TransROWL's loss function. Both TransROWL and TransROWL^R are evaluated with DBpedia15K, DBpedia100K, DBpediaYago and NELL for both link prediction (typeOf triples alone) and triple classification.

The topic is very interesting, but the technical solution seems to be only incremental. It is a (simple) combination of TransR and the existing work TransOWL [22] and the regularization [16].

The paper is easy to follow, especially the first two sections and the evaluation section. However, there are some technical issues in the methods. They are either confusing or obviously erroneous.

In the loss of TransOWL (1), why the predicate of the last term is typeOf? The corresponding triples come from $\Delta_{subClass}$. For additional triples in $\Delta_{equivClass}$ (in the third term), why it leads to typeOf triples? if h is an equivalent class to l, it should lead to <h, subClassOf, l> and <l, subClassOf, h>.
In the loss of TransROWL (2), the last term uses the predicate of subClassOf, which I think is correct. But why only the object of the original triple is considered. I mean it should also lead to <h, subClassOf, q> and <h', subClassOf, q'>, instead of just <t, subClassOf, p> and <t', subClassOf, p'>. All these need concrete and accurate descriptions to fully follow the implementation.

The loss of TransROWL (2) uses the same triple score $f_r$ as that of TransOWL (1). According to the text description, TransROWL should use the triple score of TransR (it projects the relation vector to another space by by a matrix). So I think the issue lies in the annotation. But this is really confusing. I also suggest the authors introduce TransR in comparison with TransE, in the preliminary.
Another confusing annotation is the usage of $\sum$. In Section 2 (Basics on Embedding Methods), two sums are used: one for the positive triples and one for the negative triples, while in Section 3, only one sum is used.

In the loss of TransROWL^R, what are $M_r$ and $M_p$? I guess they represent the relation projection matric. Then why the terms of $|| M_r - M_p||$ are added for both inverseOf and equivProp.

BTW, some equations have equation number while some don't.

Finally the results on typeOf link prediction and triple classification are not significantly improved w.r.t., the baselines TransR and TransOWL. They very from data to data, and from task to task. In general, both TransROWL and TransROWL^R are quite close to TransR or TransOWL.


**Anonymity:**

Yes, I would like my review to remain anonymous.

**Rating:**

-1: Weak Reject

**Reuse And Availability:**

3: Medium

**Strong Points:**

1. Some new ideas have been proposed to inject the schema's knowledge in KG embedding.
2. Extensive evaluation has been conducted on different KGs.

**Subreviewer:**

I submitted this review.

**Weak Points:**

See the review.

---

> ### Author Rebuttal · Authors · 2021-01-29
>
> “The topic is very interesting, but the technical solution seems to be only incremental. It is a (simple) combination of TransR and the existing work TransOWL [22] and the regularization [16].”
>
> __R:__ On the ground of the idea that the exploitation of expressive schema-level axioms may help when learning embeddings and consequently improve the model effectiveness, we started with proposing TransOWL and its upgraded version TransROWL, jointly with a more suitable loss function and a different scoring function. Please note that actually TransOWL is part of the paper contribution. We compressed TransOWL presentation due to lack of space, preferring to dedicate more space to TransROWL that constitutes a significant improvement of TransOWL. Indeed, as the reviewer may notice, TransOWL has not been published. We reported [22], which is actually a bachelor thesis, as a form of acknowledgement to the student working on the preliminary version of TransOWL. We have clearly understood that this resulted in some way confusing and most of all reduced the perception of novelty and advance with respect to the state of the art of our contribution. We will make this aspect absolutely clear, in the final version of the paper.
>
> “In the loss of TransOWL (1), why the predicate of the last term is typeOf? The corresponding triples come from \Delta_subClass. For additional triples in \Delta_equivClass(in the third term), why it leads to typeOf triples? if h is an equivalent class to l, it should lead to <h, subClassOf, l> and <l, subClassOf, h>.
> In the loss of TransROWL (2), the last term uses the predicate of subClassOf, which I think is correct. But why only the object of the original triple is considered. I mean it should also lead to <h, subClassOf, q> and <h', subClassOf, q'>, instead of just <t, subClassOf, p> and <t', subClassOf, p'>. All these need concrete and accurate descriptions to fully follow the implementation.”
>
> __R:__ We thank the reviewer for spotting the typos. In fact in the last term of the TransOWL equation (1) there is an error in the notation, which should follow the same format of the TransROWL equation for the case of subclassOf predicate. As for the case of the equivalentClass predicate, the rationale that has been adopted is to maximize the amount of information that we could make explicit and thus added in the considered dataset to be hence exploited during the training phase. Particularly, we grounded on the fact that the typeOf relation is one of the most common predicates in a KG, and as such we used it as a primary target of the training phase. Let us consider a class A  which has an equivalent class B. Given the triple (h,typeOf,A) we can derive also (h,typeOf,B). Training the model on those derived triples brings a higher number of typeOf triples in the considered data collection with respect to those suggested by the reviewer. We will add this explanation in the final version of the paper.
>
> “The loss of TransROWL (2) uses the same triple score as that of TransOWL (1). According to the text description, TransROWL should use the triple score of TransR (it projects the relation vector to another space by a matrix). So I think the issue lies in the annotation. But this is really confusing.”
>
> __R:__ Your observation is correct, there is an issue in the notation: the score function of TransROWL is the same as TransR because the projection matrices are involved in the calculations.
>
> “I also suggest the authors introduce TransR in comparison with TransE, in the preliminary”
>
> __R:__ we will briefly introduce TransR besides TransE when presenting basics.
>
> “Another confusing annotation is the usage of \sum. In Section 2 (Basics on Embedding Methods), two sums are used: one for the positive triples and one for the negative triples, while in Section 3, only one sum is used”
>
> __R:__ We will adopt in section 2 the same style that has been used for section 3 that is we will report  in section 2 just one \sum and the two different sets.
>
> “In the loss of TransROWL^R, what are M_r and M_p ? I guess they represent the relation projection matric. Then why the terms of ||M_r - M_p|| are added for both inverseOf and equivProp.”
>
> __R:__ Those are projection matrices, and the term ||M_r - Mp|| is necessary to impose that the matrices associated with predicates r and p must tend to be equal. This is necessary because, in order to have the same energy associated to triples like (h,r,t) and (h,p,t), were r and p are equivalentRelations, we need that the Projection Matrices M_r  and M_p must tend to be equal, that is having the same projection, and consequently the same final score function values.
>
> “BTW, some equations have equation number while some don't.”
>
> __R:__ we used numbers only when we needed to refer to equations. We will report numbers to all equations in the final paper version.

---

### Official Review · AnonReviewer3 · 2021-01-13
**Interesting approach whose presentation could be improved**

**Rating:** 1
**Confidence:** 4
**Impact:** 4
**Design And Technical Quality:** 3

**Review:**

The paper proposes a novel technique to inject background knowledge in OWL format into the $TRANSR$ embedding model. The injection of BK knowledge is performed through axiom injection in the loss function or in the regularization term. Results on link prediction and  triple classification on standard KGs are encouraging.

I found the approach very interesting even if it is not totally new: it is an extension of $TRANSOWL$ and of $TRANSE^R$. Maybe more considerations about the differences between $TRANSOWL$ and $TRANSROWL$ would improve the paper. Indeed, the differences between Eqs 1 and 2 seem to me only in the $\lambda_i$ parameters and in the last term of the loss. Is this the case? More considerations would be welcome. In addition, it is not clear to me the $R$ component of the $TRANSROWL$ method in Eq. 2. Whereas this component is clear in Eq. 3 with the difference between the $M_i$ matrices.

It would be interesting (or to discuss) the use of more kinds of axioms in Eqs. 2/3 and a rewriting of that equations in a more compact equation that ranges over the several $\Delta_{\pi}$. This would improve the readability in the case of more kinds of axioms and make the approach more general.

How did you find the $\lambda_i$ parameters? Grid search on the validation set?

What is the difference between bern and unif methods?

Tables 1 and 2 are rather dense, maybe I would put the "raw" columns as additional material. In addition, the MR columns give a different idea of the best method with respect to the H@10 ones. The same holds for P, R and acc columns. A selection of a kind of column (or more discussions about their difference) would improve the readability of the performance. Often H@10 and acc results differ of only few decimal points between the several methods, therefore it is difficult to have a clear difference between the methods. In this case cross validation would make the results stronger.

**Anonymity:**

Yes, I would like my review to remain anonymous.

**Reuse And Availability:**

4: High

**Strong Points:**

- The methods on how BK is injected into a loss function;
- the availability of source code and datasets on github;
- the evaluation on 4 standard datasets;
- in general, the paper is well written and organized even if some mentioned parts are not so clear to me and can be improved;
- the discussion about the limitations of the proposed methods.

**Subreviewer:**

I submitted this review.

**Weak Points:**

- The presentation issues in the review about Eqs. 1/2 and results;
- I would add in the introduction a brief discussion/comparison with neural symbolic methods as $TRANS(R)OWL$ can be considered neural symbolic works. In particular, Logic Tensor Networks [1] has also been applied to a type classification task in KGs and presents a way of injecting BK similar to the ones here presented.

Some typos:
- page 6, for each considered axioms -> for each considered axiom;
- page 6, to injecting -> to inject;
- page 7, withing -> within;
- page 10, perfformance -> performance;
- page 12, the the;
- "allow to" is not correct. Use "allow us to" or "allow the [NOUN]" instead.

[1] Donadello, I., & Serafini, L. (2019, July). Compensating supervision incompleteness with prior knowledge in semantic image interpretation. In 2019 International Joint Conference on Neural Networks (IJCNN) (pp. 1-8). IEEE.

---

> ### Author Rebuttal · Authors · 2021-01-29
>
> “I found the approach very interesting even if it is not totally new: it is an extension of TRANSOWL and of TRANSE. Maybe more considerations about the differences between TRANSOWL and TRANSROWL would improve the paper. Indeed, the differences between Eqs 1 and 2 seem to me only in the \lambda_i parameters and in the last term of the loss. Is this the case? More considerations would be welcome.”
>
> __R:__ On the ground of the idea that the exploitation of expressive schema-level axioms may help when learning embeddings and consequently improve the model effectiveness, we started with proposing TransOWL and its upgraded version TransROWL, jointly with a more suitable loss function and a different scoring function (being TransROWL based on TransR, it has a more effective way to represent predicates like, for example, `typeOf`). Please note that actually TransOWL is part of the paper contribution. We compressed TransOWL presentation due to lack of space, preferring to dedicate more space to TransROWL that constitutes a significant improvement of TransOWL. Indeed, as the reviewer may notice, TransOWL has not been published. We reported [22], which is actually a bachelor thesis, as a form of acknowledgement to the student working on the preliminary version of TransOWL. We have clearly understood that this resulted in some way confusing and most of all reduced the perception of novelty and advance with respect to the state of the art of our contribution. We will make this aspect absolutely clear, in the final version of the paper
>
> “In addition, it is not clear to me the R component of the method in Eq. 2. Whereas this component is clear in Eq. 3 with the difference between the M_i matrices”
>
> __R:__ We thank the reviewer for spotting the typo: the score function in Eq. 2 is the same as the one in TransR, so TransOWL and TransROWL have different formulations (since TransOWL adopt the score function proposed with TransE).
>
> “It would be interesting (or to discuss) the use of more kinds of axioms in Eqs. 2/3 and a rewriting of that equations in a more compact equation that ranges over the several \Delta_pi. This would improve the readability in the case of more kinds of axioms and make the approach more general.”
>
> __R:__ We will consider this suggestion. Thanks
>
> “How did you find the parameters?”
>
> __R:__ As for the parameter setting, for the sake of comparison, the very same values coming from the original papers proposing the baseline methods have been adopted. We will make this clear in the final version of the paper.
>
> “What is the difference between bern and unif methods?”
>
> __R:__ The difference between bern and unif methods has been formulated by the paper introducing TransH that we reported as a reference([25]). We summarize the explanation of the differences below and we will also add it in the final version of the paper. In brief, the unif generates negative triplets randomly sampling a pair of entities for head and tail from $E$ assigning uniform probabilities to the possible replacements. Whereas the bern assigns different chances (setting suitable parameters for a Bernoulli distribution) based on the nature of the mapping property (1-to-1,1-to-N, N-to-N) of the underlying relation.
>
> “Tables 1 and 2 are rather dense, maybe I would put the "raw" columns as additional material. In addition, the MR columns give a different idea of the best method with respect to the H@10 ones. The same holds for P, R and acc columns. A selection of a kind of column (or more discussions about their difference) would improve the readability of the performance.”
>
> __R:__ We will consider this suggestion. Thanks
>
> “I would add in the introduction a brief discussion/comparison with neural symbolic methods as can be considered neural symbolic works. In particular, Logic Tensor Networks [1] has also been applied to a type classification task in KGs and presents a way of injecting BK similar to the ones here presented.”
>
> __R:__ We will consider this suggestion in the preparation of the revised version. Thanks

---

### Official Review · AnonReviewer1 · 2021-01-14
**Consistent with previous work**

**Rating:** 1
**Confidence:** 4
**Impact:** 3
**Design And Technical Quality:** 3

**Review:**

This paper delves into the transE and transR families of knowledge graph embeddings, proposing two different approaches to extend TRANSOWL, which exploits available background knowledge based on the transE setting, resorting to schema axioms to derive new triples that can be used during training. The extensions to TRANSOWL proposed in this paper, called TRANSROWL and TRANSROWLR, take inspiration in the analogous steps taken back in the day when transR was proposed as a way to address the weak points of transE, including the modeling of specific types of relationships.

While TRANSROWL leverages properties like inverseOf, equivProperty and subClass to explicitly generate triples used for training, TRANSROWLR exploits background knowledge by making changes in the loss function, following the work by Minervini et al. (2017) to take into account the regularization of the embeddings.

The paper is quite predictable in the sense that it logically derives from previous work in knowledge graph embeddings. The work is interesting although not particularly innovative. The algorithms are properly formulated and duly justified. The evaluation is sufficient, focused on the link prediction and triple classification tasks, which are used to evaluate the algorithms  proposed over three datasets (DBpedia, YAGO and NELL). The datasets are also the same used in previous related work. On one hand, this is OK since it is necessary to compare the algorithms proposed against the state of the art. On the other hand,  it would have been interesting to evaluate also against some additional dataset to illustrate the benefits of the approach in a new setting.

For DBPedia15K, it seems TRANSROWL is marginally better than TRANSR in link prediction for the typeOf property. It is also close, but worse than TRANSR in other properties. TRANSROWLR is generally worse (not by much) , except for H@10 in typeOf. This seems to change in favor of TRANSROWLR with a larger fragment of DBPedia (100K). Similarly in DBPediaYAGO, although in NELL TRANSR is generally the best for no TypeOf properties and on par with TRANSROWL otherwise. Similarly, for triple classification, the results seem to occasionally benefit the proposed algorithms as the number of triples in the dataset grows. In both tasks, NELL seems to be the more difficult dataset for TRANSROWL and TRANSROWLR, which seems to confirm a correlation with the completeness of the graph.

In terms of writing style, the paper tends to fall on the long-winded side. It takes a long time until you get to the actual contributions, after a long introduction to knowledge graph embeddings that is probably unnecessary given the proper references are in place. The contributions themselves are not explicitly stated in a clear way and kind of merge with the algorithms that are being extended. Finally, the paper could benefit from some conciseness, which would improve readability, as well as from a double check on the English.




**Anonymity:**

Yes, I would like my review to remain anonymous.

**Reuse And Availability:**

5: Very High

**Strong Points:**

- Interesting extension of well known algorithms.
- Sufficiently evaluated.
- GitHub repo available.

**Subreviewer:**

I submitted this review.

**Weak Points:**

- Logical evolution of pre-existing approaches, limited innovation.
- Sometimes long-winded and hard to read.

---

> ### Author Rebuttal · Authors · 2021-01-29
>
> “The contributions themselves are not explicitly stated in a clear way and kind of merge with the algorithms that are being extended.”
>
> __R:__ On the ground of the idea that the exploitation of expressive schema-level axioms may help when learning embeddings and consequently improve the model effectiveness, we started with proposing TransOWL and its upgraded version TransROWL, jointly with a more suitable loss function and a different scoring function. Please note that actually TransOWL is part of the paper contribution. We compressed the presentation of TransOWL due to lack of space, preferring to dedicate more space to TransROWL that constitutes a significant improvement of TransOWL. Indeed, as the reviewer may notice, TransOWL has not been published. We reported [22], which is actually a bachelor thesis, as a form of acknowledgement to the student working on the preliminary version of TransOWL. We have clearly understood that this resulted in some way confusing and most of all reduced the perception of novelty and advance with respect to the state of the art of our contribution. We will make this aspect absolutely clear, in the final version of the paper, by saving space from the introductory and background sections which will be made shorter, as suggested by the reviewer.
>
> “The datasets are also the same used in previous related work. On one hand, this is OK since it is necessary to compare the algorithms proposed against the state of the art. On the other hand, it would have been interesting to evaluate also against some additional dataset to illustrate the benefits of the approach in a new setting.”
>
> __R:__ our main aim was the comparison with the target methods where the datasets that we used for our evaluation have been adopted. Experiments on additional datasets are part of our ongoing work.

---

> > ### Comment · AnonReviewer1 · 2021-02-03
> > **Post-rebuttal**
> >
> > Thank you for your clarifications. Indeed, trying to make the contribution proposed clearer in the context of previous work would help to better assess its novelty. I would reduce the intro and dedicate more space to single out your work.

---

### Official Review · AnonReviewer5 · 2021-01-14
**Good elaboration of background and presentation of method. Evaluation lacking.**

**Confidence:** 3
**Impact:** 3
**Design And Technical Quality:** 4

**Review:**

This work improves existing KG embedding models by injecting background knowledge by modifying the loss function to specifically consider selected properties.

The work gives a good introduction of the background required to understand the method and explains the motivation behind the approach well. The mathematical notation is mostly clear, even though it could have been more concise.

The main shortcomings of the work is the experimental section.

Experimental setup:
* Without tuning the learning rate the model's performance is likely to be sub-optimal.
* The selection procedure for the method-specific parameters is unclear. For example, lambda 1 through 6 are set to 0.1 in TransRowlR. Without further explanation, this seems arbitrary.
* It's unclear why the authors choose to show different number of epochs for some methods (e.g, TransR) but not for others (e.g., TransROWL). Preferably, only the best performing model could be shown to make experiments more concise.
* In the table 1 and 2 it's hard to distinguish the baselines from the proposed method.

Experimental results:
* The evaluation of the method seems a bit hand-wavy "...in most of the cases scores better than the base model." A deeper analysis or explanation for the behavior would have been beneficial.
* The improvement of the proposed method is often marginal. Significance tests would strengthen the argument here.
* Compare to more recent SOTA baselines, e.g., Nie and Sun 2018: "Joint knowledge base embedding with neighborhood context"

**Anonymity:**

Yes, I would like my review to remain anonymous.

**Rating:**

-1: Weak Reject

**Reuse And Availability:**

4: High

**Subreviewer:**

I submitted this review.

---

> ### Author Rebuttal · Authors · 2021-01-29
>
> “Experimental setup”
>
> __R:__ As for the parameter setting, for the sake of comparison, the very same values coming from the original papers have been adopted. We will make this clear in the final version of the paper.
>
> “It's unclear why the authors choose to show different number of epochs for some methods (e.g, TransR) but not for others (e.g., TransROWL).”
>
> __R:__ We assume the reviewer is referring to the fact that 1000 epochs have been used for TransE and TransOWL whilst for all the other methods that are somehow related to TransR, 250 and 500 epochs have been also adopted besides 1000.  As reported at the beginning of section 4.1 (Parameter Setting), the motivation is given by the fact that TransR requires an initialization of the embeddings performed via TransE (due to a tendency to overfitting that is known to affect TransR more). Similarly, the variants of TransROWL were initialized with embeddings computed using TransE. After the initialization with embeddings found by TransE, hypothesizing that overfitting may affect the other models derived from TransR, they have been trained with increasing numbers of epochs to check the occurrence of this phenomenon. We will try to make this even more explicit in the final version of the paper.
>
> “In the table 1 and 2 it's hard to distinguish the baselines from the proposed method.”
>
> __R:__ We will report the baseline in bold to make it clear in the table, besides in the text.
> “Experimental results: The evaluation of the method seems a bit hand-wavy "...in most of the cases scores better than the base model." A deeper analysis or explanation for the behavior would have been beneficial.”
>
> __R:__ we will try to extend the discussion of the experimental results a bit more.
>
> “Compare to more recent SOTA baselines, e.g., Nie and Sun 2018: "Joint knowledge base embedding with neighborhood context" “
>
> __R:__ The proposed research direction is in principle applicable to other embedding methods but some additional work is needed to come up with a proper formalization e.g. of the corresponding loss function (that is anyhow possible). Our main aim here was to show the feasibility of the approach, starting with well known methods before moving on towards more complex (and recent) solutions.

---

> > ### Comment · AnonReviewer5 · 2021-02-08
> > **Acknowledged**
> >
> > Thanks!

---

### Decision · Program_Chairs · 2021-02-23

**Decision:**

Accept with shepherding

**Comment:**

The paper addresses an important problem of injecting background knowledge into embedding models, and it certainly fits the ML track of ESWC. However, there are a few issues raised by reviewers that need to be addressed before its acceptance. As we believe that this is doable given the time before the camera-ready deadline, conditional acceptance is suggested. The paper can only be accepted if these issues have been addressed.

Below we provide the list of issues

1. Expansion of the evaluation:
    * Experiments on more complex ontologies;
    * Discussion on the behavior of the method.
2. Discussion of the related work:
    * Extensive comparison with existing state-of-the-art neural-symbolic methods.  Incorporating the schema in embedding models has been investigated quite a bit, e.g., in
        * https://www.aclweb.org/anthology/D16-1019/,
        * https://arxiv.org/abs/1707.07596,
        * https://arxiv.org/abs/1805.10461,
        * https://arxiv.org/abs/2007.06267
3. Readability
    * Making contributions more distinct